# Effect of Domain Structure and Dielectric Interlayer on Switching Speed of Ferroelectric Hf_0.5_Zr_0.5_O_2_ Film

**DOI:** 10.3390/nano13233063

**Published:** 2023-12-01

**Authors:** Anastasia Chouprik, Ekaterina Savelyeva, Evgeny Korostylev, Ekaterina Kondratyuk, Sergey Zarubin, Nikita Sizykh, Maksim Zhuk, Andrei Zenkevich, Andrey M. Markeev, Oleg Kondratev, Sergey Yakunin

**Affiliations:** 1Moscow Institute of Physics and Technology, National Research University, Institutskii per. 9, 141701 Dolgoprudny, Russia; saveleva.eo@mipt.ru (E.S.); korostylev.ev@phystech.edu (E.K.); ekaterina.v.kondratyuk@phystech.edu (E.K.); zarubin.ss@mipt.ru (S.Z.); sizykh.na@phystech.edu (N.S.); maksimzhu1994@yandex.ru (M.Z.); zenkevich.av@mipt.ru (A.Z.); markeev.am@mipt.ru (A.M.M.); 2National Research Center “Kurchatov Institute”, 123098 Moscow, Russia; kondratev_oa@nrcki.ru (O.K.); s.n.yakunin@gmail.com (S.Y.)

**Keywords:** ferroelectric memory, ferroelectric hafnium oxide, ferroelectric thin film, polarization switching kinetics, domain structure

## Abstract

The nanosecond speed of information writing and reading is recognized as one of the main advantages of next-generation non-volatile ferroelectric memory based on hafnium oxide thin films. However, the kinetics of polarization switching in this material have a complex nature, and despite the high speed of internal switching, the real speed can deteriorate significantly due to various external reasons. In this work, we reveal that the domain structure and the dielectric layer formed at the electrode interface contribute significantly to the polarization switching speed of 10 nm thick Hf_0.5_Zr_0.5_O_2_ (HZO) film. The mechanism of speed degradation is related to the generation of charged defects in the film which accompany the formation of the interfacial dielectric layer during oxidization of the electrode. Such defects are pinning centers that prevent domain propagation upon polarization switching. To clarify this issue, we fabricate two types of similar W/HZO/TiN capacitor structures, differing only in the thickness of the electrode interlayer, and compare their ferroelectric (including local ferroelectric), dielectric, structural (including microstructural), chemical, and morphological properties, which are comprehensively investigated using several advanced techniques, in particular, hard X-ray photoelectron spectroscopy, high-resolution transmission electron microscopy, energy dispersive X-ray spectroscopy, X-ray diffraction, and electron beam induced current technique.

## 1. Introduction

The discovery of ferroelectricity in doped hafnium oxide (HfO_2_)-based thin films has drawn significant attention because of their inherent advantages over perovskite ferroelectrics related to CMOS compatibility and high scalability [1,2,3]. To date, many non-volatile memory devices based on ferroelectric HfO_2_ films with excellent performance have already been demonstrated. Despite this progress, some challenges have still not been overcome and they hinder the commercialization of HfO_2_-based memory. In particular, HfO_2_ films demonstrate a relatively low polarization switching speed, which is about two orders of magnitude lower than in perovskite films [4]. Meanwhile, switching speed limits the operating frequency and operating voltage of the memory chips.

Polarization switching speed and related domain dynamics are important characteristics for other applications of ferroelectric films as well as, for example, in-memory computing. This direction includes in-memory ferroelectric logics [5,6,7] and neuromorphic computing [8,9]. The neuromorphism of ferroelectric structures stems from the multiple states provided by varying domain configurations. Different neuromorphic applications demand a wide range of operating frequencies—from hundred Hz for the emulation of biological synapses [10,11,12] to THz for the ultrafast processing of data and reduced energy consumption [13]. The design and implementation of such devices requires an understanding of both the intrinsic kinetics of polarization switching and the external mechanisms that lead to its degradation in real devices.

The exact reason for the low polarization switching speed in HfO_2_ films is still unknown. The effect of various parameters of HfO_2_ on its switching rate has been investigated. In particular, it has been shown that HfO_2_ film thickness [14], concentration of oxygen vacancies [15,16], and Zr alloy concentration in Hf_1−x_Zr_x_O_2_ thin films [17] affect the polarization switching kinetics. On the other hand, the impact of several physical phenomena in HfO_2_ on time dependence of polarization switching has been examined. In particular, it has been shown that it depends on local field inhomogeneity [17] and charge injection into functional HZO layer [18].

In addition to HfO_2_-specific factors, the polycrystallinity of the material and the fundamental properties of ferroelectrics influence the switching speed. For other ferroelectric polycrystalline films, it has been shown that the grain pattern (e.g., grain size and orientation) and grain boundaries affect the polarization switching or domain evolution dynamics in ferroelectric nanofilms [19,20]. Furthermore, the depolarization field of spontaneous polarization, an intrinsic feature of ferroelectrics, has been reported to be the cause of the slowing down of polarization switching [21,22]. Interestingly, the influence of the depolarization field differs for hafnium oxide and perovskites [21,22]. This is probably due to the different film thicknesses and different concentrations of charged defects that screen the depolarization field [23].

Therefore, polarization reversal in ferroelectric HfO_2_ film has a complex nature, and polarization switching kinetics is determined by a number of factors and their interplay. In this work, we show that the domain structure and the interfacial dielectric layer influence each other and together contribute significantly to the polarization switching speed. More specifically, the dielectric layer is formed at the electrode interface due to the reduction of hafnium oxide, which changes the ferroelectric properties of the film. On the one hand, the internal field in the ferroelectric decreases and the coercive voltage increases, and on the other hand, charged defects are formed in the film bulk. The first phenomenon reduces the polarization switching speed due to the fundamental laws of polarization switching kinetics, whereas charged defects pinch domain walls during domain propagation and reduce the domain size. In order to resolve this issue, two types of W/Hf_0.5_Zr_0.5_O_2_ (10 nm)/TiN structures consisting of identical functional materials but slightly different dielectric WO_x_ interlayer thicknesses are fabricated. Their structural (including microstructural), chemical, morphological properties are comprehensively investigated using high-resolution transmission electron microscopy, energy dispersive X-ray spectroscopy, hard X-ray photoelectron spectroscopy, X-ray diffraction, electron beam induced current technique, scanning electron microscopy, and atomic force microscopy. Based on the comparison of the properties of the structures, the influence of interlayer and domain structure on the switching speed is elucidated. The results could be useful for the material and interface engineering required for the development of the next-generation non-volatile ferroelectric memories.

## 2. Materials and Methods

*Structures fabrication*. Hf_0.5_Zr_0.5_O_2_ (HZO) films 10 nm in thickness were grown on preliminarily W-electroded silicon (100) substrates via thermal atomic layer deposition at a 240 °C substrate temperature using Hf[N(CH_3_)(C_2_H_5_)]_4_ (TEMAH), Zr[N(CH_3_)(C_2_H_5_)]_4_ (TEMAZ) and H_2_O as precursors and N_2_ as a carrier and a purging gas. A 40 nm thick W film and TiN (20 nm) films deposited by magnetron sputtering served as the bottom and top electrodes, respectively. To fabricate structures with different dielectric WO_3_ interlayers, after the preparation of the bottom W electrode, HZO growth substrates of type No. 1 were first slightly oxidized by heating at 50 °C in water vapor (85% humidity) for 1 h. The samples of type No. 2 were in room conditions at this time. To minimize *RC* time constant through minimizing of contact resistance, top TiN electrodes were covered by an Al layer (150 nm) using electron beam evaporation. Patterning of the top electrode was performed using maskless optical lithography and plasma chemical etching. Crystallization of HZO film was induced by rapid thermal annealing at 500 °C for 30 s in an Ar atmosphere. 

For the in situ EBIC study, the functional capacitors are routed to the Al contact pads, allowing for external electric biasing of the capacitors. The routing fabrication details have been described previously [24].

For *ferroelectric characterization*, the Cascade probe station coupled with the semiconductor parameter analyzer B1500A (Agilent Technologies, Santa Clara, CA, USA) was used. Switching *I-V* and *P-V* curves were measured through the dynamic positive-up negative-down (PUND)-like technique [25]. To wake up the as-prepared HZO film, the ferroelectric capacitors were cycled 10^5^ times by applying bipolar voltage double triangular pulses with a duration of 10 µs and an amplitude of ±3 MV/cm. The studies were carried out using ferroelectric HZO–based capacitors 25 μm in size. The *RC* time constant was less than 30 ns.

*Capacitance transient spectroscopy* was conducted using the 50 MHz lock-in amplifier HF2LI (Zurich Instruments, Zurich, Switzerland) and current amplifier HF2TA (Zurich Instruments, Zurich, Switzerland). Preliminary calibration was performed to eliminate the contribution of the impedance of the measuring setup. The amplifier signal had a zero DC bias, an effective AC voltage of 100 mV, and a frequency of 100 kHz. The voltage pulse with a duration of 5 s and amplitude of 3 V was used for the saturation of traps in HZO layer.

*Atomic force microscopy (AFM)*. Topography maps of the surface of HZO film were acquired using an atomic force microscope AFM Ntegra (NT-MDT Spectrum Instruments, Moscow, Russia) operated in tapping mode. Composite poly-Si&Si cantilevers HA_FM Etalon (ScanSens, Bremen, Germany) with a free resonance frequency of 80 kHz and a force constant of 3.5 N/m were used.

*Scanning electron microscopy (SEM)*. For the characterization of grain size, images of HZO films surfaces were acquired using a scanning electron microscope JEOL JSM-7001F (JEOL, Tokyo, Japan) operated in secondary electrons mode with a primary electron beam energy of 30 keV.

*Electron beam-induced current (EBIC) technique*. By EBIC technique, the domain structure was mapped in situ during polarization switching. The scanning electron microscopy setup (JEOL JSM-7001F, JEOL, Japan) was used to provide the focused electron beam, which induced current in the closed-loop electric circuit containing the capacitor device under investigation. To measure the induced current, a home-made holder with amplifying circuit was used. The amplitude of flowing current was defined by the local uncompensated depolarization electric field, pulling apart the electron-hole pairs generated a due to the interaction of the electron beam with the sample. Scanning the beam across the surface allowed for the obtaining of the map of spatial distribution of the depolarization field in the thin film. Since the depolarization field is co-directional with the polarization vector, the depolarization field map is related to the domain structure and, therefore, the polarization state and domain size can be determined. The polarization switching was performed by trapezoidal voltage pulses of alternating amplitudes in the range from −3 V to 3 V and with 1 ms duration, supplied by waveform generator Rigol DP5352 (Rigol Technologies, Inc., Suzhou, China). Moreover, additional images of sample surface were acquired by SEM in back-scattered electrons mode in order to compensate drift of beam focus and sample position. A detailed description of device fabrication and basic experimental procedure has been described earlier [24]. All measurements were performed with awakened capacitors, which ensured that any structural transformations possible during the initial stage of capacitor operation were ended.

*High-Resolution Transmission Electron Microscopy (HRTEM) and Energy-Dispersive X-ray Spectroscopy (EDX)* study was conducted on a JEM-2100 (Jeol, Tokyo, Japan) transmission electron microscope (TEM) operated at an accelerating voltage of 200 kV. The TEM was equipped with an X-Max^N^ 100 TLE Silicon Drift Detector for EDX analysis. Cross-section lamellas were prepared by focused ion beam milling after conducting the wake-up procedure to ensure any possible structural transformations were ended.

*X-ray diffraction (XRD).* The analysis of the structural phase composition of the ferroelectric film was performed using XRD patterns obtained using a Rigaku SmartLab 9 kW diffractometer (Rigaku, Tokyo, Japan) with a rotating Cu-anode by 2θ (detector) scans, while the angle between the sample surface and the incident beam was 0.65 degrees. That angle was chosen as the best in terms of the signal-to-noise ratio. The incident beam was collimated by a double-crystal Ge (220) monochromator to suppress the Kα_2_ and Kβ lines of the characteristic Cu K-lines. In addition, there were slits in the diffraction plane, perpendicular to it, with sizes of 0.1 mm and 5 mm to set the studied area of the samples. The slit before the detector was 0.5 mm to cut off the background radiation. The 2θ step of the obtained X-ray patterns was 0.025 degrees within a 27–38 degrees area, with a time per point of 300 s.

*Hard X-ray Photoelectron Spectroscopy (HAXPES)*. To elucidate depth-dependent differences, the bilayer model analogs of both samples were examined by HAXPES under near-total reflection condition (NTR-HAXPES) [25,26,27]. Usually, X-ray photoelectron spectroscopy (XPS) is employed to reveal the chemical state of a (sub) surface layer of a material, but it can also be applied to probe the electronic properties of a dielectric layer. In particular, charged defects present in a dielectric give rise to the potential distribution across the layer, thus affecting the energy of photoelectrons and ultimately the core-level spectra of dielectric constituents. Here, it is worth noting that while neutral oxygen vacancies at the concentrations of 10^18^–10^19^ cm^−3^ are below the sensitivity of XPS, charged oxygen vacancies manifest themselves by broadening and shifting the dielectric core-level lines. However, in laboratory XPS with X-rays at an energy of about 1 keV, the photoelectrons have quite a small inelastic mean free path (IMFP), which translates to a limited (3–5 nm) probing depth. In order to overcome this limitation, one can use hard X-rays (>5 keV), yielding much larger probing depths (up to 20 nm). Yet, it comes at the cost of a dramatic reduction in the photoelectron emission cross-section and, therefore, the use of (extremely bright) synchrotron radiation sources is preferable. Moreover, by irradiating the sample surface mounted on a high precision manipulator with a monochromatic focused X-ray at the glancing angles, one can achieve a stationary X-ray wavefield in the irradiated sample configured as a standing-wave. The position of the X-ray standing wave intensity maximum in this case strongly depended on the incident angle, and by changing the latter one could precisely localize the region where the photoelectrons come from. HAXPES measurements were performed (beamline P22 of PETRA III, DESY, Gamburg, Germany) at an excitation energy of about 6 keV, with Specs 225 HV used as an analyzer. The glancing angles of incidence were varied in the range from 0.2 to 1.1 degrees (with an angle increment of 0.025 and 0.040 degree), with nearly normal photoelectron take-off geometry. Hf4f and Zr3p were acquired as the main spectral regions of interest. NTR rocking curves (RC; core-level signal intensity vs. angle of incidence) were obtained based on accurate spectra fitting with Voigt function lineshape and Shirley background. Experimental NTR RC were matched with the sample structure-based photoelectron yield calculation [28,29,30] implemented on the top of TER_sl program [31]. The fitting of NTR rocking curves using the roughness of HZO layer and a possible interlayer at W/HZO interface as free model parameters (while setting HZO thickness to 10 nm as determined by TEM analysis) allowed for us to elucidate the presence of WO_x_ and to precisely determine its thickness.

## 3. Results and Discussion

First of all, let us analyze what the difference is in the functional properties of W/HZO/TiN ferroelectric capacitors, which are almost identical and only slightly differ by the state of the bottom electrode modified before HZO film growth. Let us call structures with a slightly oxidized W bottom electrode “structures of type No. 1” and those with an unmodified W electrode “structures of type No. 2” (Figure 1a).

Comparing the ferroelectric properties of the structures measured after the wake-up procedure (Figure 1b,c), one can conclude that the structures of type No. 1 show better performance. Namely, they have higher remanent polarization: 26 μC/cm^2^, compared to 20 μC/cm^2^ for the structures of type No. 2, and lower coercive voltage: 1.1 V and −1 V, compared to 1.3 V and −1.2 V for the structures of type No. 2 (Figure 1b,c). This would potentially allow for a slightly larger memory window and a slightly lower write/read voltage, i.e., lower power consumption. On the other hand, it should be noted that the leakage current is an order of magnitude larger (Figure 1d), which affects power consumption in the opposite way. 

In order to obtain an initial idea of the reasons for the difference in ferroelectric properties, the switching *I-V* and *P-V* curves of the as-prepared structures can be analyzed. The very first two curves reflect the state of the as-prepared structure, i.e., whether the HZO film becomes mono- or polydomain right after crystallization. Indeed, obvious splitting of the switching *I-V* curve and related pinched *P-V* curve observed for the structure of type No. 2 (Figure 1f,g) indicate the existence of two opposite internal built-in fields, which are signatures of the polydomain structure in the as-prepared film [32,33]. The polydomain structure that is peculiar to the structure of type No. 2 is caused by the charged oxygen vacancies formed at both bottom and top electrode interfaces during annealing because of redox reactions with the electrode materials [3]. Depending on the mutual distribution of oxygen vacancies at the two interfaces during crystallization into the ferroelectric phase, their sum electric field locally switches the HZO film to either upward or downward polarization. Immediately after local polarization switching, the emergence of built-in electric fields and their gradual increase with time are observed. This phenomenon is caused by the redistribution of non-ferroelectric charges under the action of the depolarization field, which is aligned with the polarization vector. As a result, two opposite built-in fields appear in the domains of two types, and they manifest themselves in the splitting of the *I-V* curve and the pinching of the *P-V* curve.

For the structure of type No. 1, splitting of the switching *I-V* curve and pinching of *P-V* curve are much smaller. Instead, the *P-V* curve is shifted towards positive voltages, indicating a nearly monodomain state of the as-prepared structure and hence a unidirectional internal field built-in in the HZO film. This is probably due to the absence of oxygen vacancies at one of the electrode interfaces, i.e., the chemical inertness of one of the electrodes. A possible reason could be the existence of a chemically inert dielectric layer WO_3_ formed at the lower interface before the growth of the HZO film.

During multiple switching by bipolar voltage pulses (so called *wake-up procedure*), charged oxygen vacancies redistribute from the electrode interfaces into the film bulk. As a result, the built-in fields disappear, the *I-V* and *P-V* curves become non-split and open, respectively, the switchable polarization increases (Figure 1e), and the ferroelectric properties stabilize [25]. After the wake-up procedure, structures of both types show large remanent polarization and small coercive voltages, i.e., by these parameters, they are suitable for the implementation of ferroelectric memory.

However, a significant difference in the polarization switching speed is observed. Figure 2b shows the switching kinetics datasets measured with a conventional pulse sequence (Figure 2a), which indicate worse kinetics of polarization switching in the structures of type No. 2.

In the first step, one could compare experimental switching speed. Since the coercive voltages are different, for comparison, we choose several families of points that have the same normalized polarization at the longest duration of set pulse *t*_set_ (Figure 2c). For every pair of curves, it can be seen that when moving towards small *t*_set_, the read polarization decreases faster for samples of type No. 1, that is, at small read times (high operating frequency) the read polarization is smaller. This means that the switching time distributions for samples of type No. 1 are narrower on average. It is worth noting that at first glance, it may seem that the switching speeds for samples of type No. 1 are smaller, but this is not the case since in Figure 2c we analyze polarization switched by different *V*_set_.

In the second step, the experimental data are processed using a model of nucleation-limited switching (NLS) [34]. In the NLS model, the ferroelectric film consists of numerous areas (domains), each with their own independent switching time. The time needed for the switching of an elementary region (switching time) is equal to the waiting time for the nucleation of a reversed domain. For polycrystalline ferroelectric films, switching times are distributed in an exponentially wide time interval. According to this model, the fraction of the switched ferroelectric film *P*(*t*) is related with the distribution function of switching times *g*(*z*) by the following expression [34]:(1)$$P\left(t\right)=\int_{-\infty }^{\ln t}g\left(z\right)\;dz,$$
where $P\left(t\right)=1-\frac{\sum_{NS}S_{i}}{\sum_{all}S_{i}}$ (*NS* means non-switched domains) and the distribution function meets the normalizing condition $\int_{-\infty }^{\infty }g\left(z\right)dz=1$, *z* = log *t*_0_. Here *S*_i_ is area of *i*-th domain and *t*_0_ is the unique waiting time of the elementary regions.

In the classical application of the NLS model, its authors propose the use of a distribution function of the switching times, which is flat for *z* lying between *z*_1_ and *z*_2_ ($t_{\mathrm{left}}=10^{z_{1}}$ and $t_{\mathrm{right}}=10^{z_{2}}$ stand for the left and right limits of the flat part of the switching times spectrum):(2)$$g\left(z\right)=\frac{\Gamma^{2}h}{{\left(z-z_{1}\right)}^{2}+\Gamma^{2}}\;\;\;\mathrm{at}\;z\;<z_{1},\;g\left(z\right)=h\;\;\;\mathrm{at}\;\;z_{1}<z<z_{2},\;g\left(z\right)=\frac{\Gamma^{2}h}{{\left(z-z_{2}\right)}^{2}+\Gamma^{2}}\;\;\;\mathrm{at}\;\;z_{2}<z,\;h={\left(z_{2}-z_{1}+\Gamma \pi \right)}^{-1}.$$

Here *h* and Γ are fitting parameters for the amplitude and the rate of decay of the distribution function, respectively.

Fitting function (2) produces linear dependence of the switched polarization on log *t*_set_ in a medium range of polarization values. Full dependence of the switched polarization on log *t*_set_ is similar to the *arctan* function (solid lines in Figure 2b). Previously, it has been shown in HZO film fitting that parameter Γ is much larger than for perovskite ferroelectrics, while the flat area of time distribution is short [21]. This observation is valid for studied structures of both types. However, for type No. 2, the switching time distribution is significantly broadened, especially for small *V*_set_. The broadening of the distribution reaches three orders of magnitude of time. With equal coercive voltages, this would mean that part of the film No. 2 switches faster and part slower than it does in the film No. 1. Taking into account that the coercive voltages of the film No. 2 are larger, one could conclude that film No. 1 simply switches 1–3 orders of magnitude faster, depending on the applied voltage. This means a large degradation of the operating frequency of the memory chip.

The largest difference in switching speed is observed for low read voltages *V*_set_. It may seem that these low *V*_set_ values are far from operating voltages (~3 V). However, during long-term information storage in poled ferroelectric capacitors, an internal built-in field emerges and gradually increases with time (so-called *imprint effect* [35,36]). During the readout procedure, the built-in field partially offsets the applied field and, thus, real field in ferroelectrics turns out to be smaller than the expected field. This results in a reduction in the switching speed, which could cause readout failure.

The observed difference in ferroelectric properties and switching speed can originate from several possible causes, including structural phase composition, properties of the electrode interface, and density of defects in HZO film. Let us consider their contributions.

First of all, by analysis of the XRD patterns, we compare the structural phase composition of structures No. 1 and 2. The HZO film is a polymorphic material and may contain a mixture of ferroelectric orthorhombic and non-ferroelectric monoclinic phase grains. Indeed, as-deposited HfO_2_ film is amorphous, and ferroelectricity is induced by film crystallization during a subsequent annealing. Hafnium oxide has several structural polymorphs, and its equilibrium structural phase is the dielectric monoclinic *P*2_1_/*c* phase; whereas, the ferroelectric orthorhombic *Pca*2_1_ phase is metastable [3]. After crystallization, the HfO_2_ film becomes polycrystalline, and the ferroelectric properties strongly depend on the fraction of ferroelectric grains and their texture. In particular, this significantly affects the measured remanent polarization value. 

Figure 3 shows that the fraction of the non-ferroelectric phase is negligible in both samples and thus cannot be the cause of the different remanent polarization. The only difference in the XRD patterns is the ratio of intensities of peaks 002 and 111 of the orthorhombic ferroelectric phase, which is larger for film No. 1. Peak 002 corresponds to the vertical orientation of the polar axis, while peak 111 corresponds to the polar axis oriented at an angle of 57.7° to the film plane. As was shown earlier [37], a larger ratio of the intensities of peaks 002 and 111 correlate with a larger measured remanent polarization. In the current work, a correlation between a larger ratio of peak intensities and a larger remanent polarization is also observed (Figure 1c).

The more vertical component of the polarization vector of film No. 1 may also result in a lower value of measured coercive voltage. Indeed, if the polar axis is oriented vertically, the polarization reversal takes place with a smaller applied field. This could explain the different values of the coercive voltages, but this difference may arise from other causes as well.

To gain more insight into the structural and chemical properties of the capacitor structure that may influence their ferroelectric properties, we study their electrode interfaces using HRTEM, EDS, and HAXPES. TEM images (Figure 4a) of the cross-sections do not reveal obvious difference of the W/HZO interfaces between the samples. Measurements of the interface roughness conducted on multiple images show mean RMS roughness 1.06 and 0.99 nm for the structures of type No. 1 and 2, respectively.

An intermediate layer (interlayer) at the W interface can have a large influence on the measured ferroelectric quantities. Therefore, we paid much attention to the search for the WO_3_ (or WO_x_) layer in both samples and its characterization. This layer turned out to be indistinguishable in the HRTEM images, so we performed EDX studies.

To estimate the difference in mean WO_x_ thicknesses between the samples, several EDX profiles were obtained in several regions and analyzed for each cross-section. As an example, W and O profiles and a corresponding region of the sample can be seen in Figure 4b–e. For each set of EDX profiles of W and O for each region, the value *h* = *x*(W) − *x*(O) was calculated. Here, *x*(M) is the coordinate of the point in the EDX profile for an element, M, where intensity is an average between low and high levels of EDX signal near the W/HZO interface (*x* axis is codirectional with structure growth direction). This quantity should correlate with the WO*_x_* thickness. Average values of *h* = −0.46 nm (RMS of 0.6 nm) and *h* = −0.25 nm (RMS of 1 nm) were found for structures of type No. 1 and 2, respectively. Thus, the difference in *h* values is small and can be attributed to the modification of the W interface of structure No. 1 prior to HZO growth. Therefore, analysis of HRTEM images and EDX profiles does not reveal a significant difference between the samples’ cross-sections.

In order to investigate the W/HZO interface in more detail, we employed the HAXPES technique, which allows for the probing of non-destructively down to 20 nm into the structure. First of all, for both samples, Hf4*f* and Zr3*p* core-level spectra obtained over the entire angular range could be fitted with only one narrow doublet spectral component (Figure 5a), thus ruling out any off-stoichometry in HZO within the sensitivity of XPS technique.

Furthermore, since the NTR HAXPES technique is extremely sensitive to the presence of ultrathin (down to sub-nm) interfacial or surface oxides, it can be used to elucidate the presence of WO_x_ at the bottom W/HZO interface. By fitting experimental angular dependent intensity modulation curves (“rocking curves”) (Figure 4c,e) with simulated X-ray angular dependence maps (Figure 5b,d), we reveal the presence of WO_x_ layers at the W/HZO interface of 1 and 2 nm in thickness for samples No. 1 and 2, respectively (Figure 5b–e). In accordance with Kraut’s method [38], conduction electronic band offsets (CBO) were reconstructed at the W/HZO interfaces for both samples and surprisingly yielded the same value, with discrepancy not exceeding 0.035 eV.

Thus, it was found that the WO_x_ interlayer thickness in the two samples is different and equals to 1 and 2 nm for structures of type No. 1 and 2, respectively. It is remarkable that the WO_x_ interlayer is thicker in sample No. 2 with the unmodified W bottom electrode. This is probably due to the chemical activity of as-prepared tungsten. It is known that transition oxide films’ oxide enters into a chemical reduction–oxidation (redox) reaction with the electrode materials, resulting in the oxidation of the electrode and the reduction of the oxide at the interface [39,40]. In structures of type No. 1, the interlayer formed prior to HZO film growth may be a fairly stoichiometric oxide WO_3_ and serve as a barrier to further oxidation of the electrode. Anyway, the difference in the thickness of the interlayers is too small to have a significant effect on the measured ferroelectric quantities. However, let us take a closer look at this issue.

First, the W/WO_x_/HZO/TiN structure can be represented as two capacitors connected in series. This circuit works as an AC voltage divider; meanwhile, namely, AC voltage is used in the PUND technique. During the measurement of the *P-V* curve, the voltage in the ferroelectric layer appears to be smaller than the applied voltage and is smaller the thicker the dielectric interfacial WO_x_ layer. Therefore, a larger external voltage is needed for the structures of type No. 2 to achieve the coercive field than for the structures of type No. 1. In Figure 1c, it is the applied voltage that is plotted along the horizontal voltage axis. We call the applied voltage at which polarization switching is observed the coercive voltage. Therefore, the measured coercive voltage is higher for the structures of type No. 2 compared to the structures of type No. 1.

Second, it is known that ferroelectricity in hafnium oxide films is promoted by using substrates and electrode materials with a low thermal coefficient of expansion (TCE) compared to the TCE of hafnia. In this case, during annealing, the hafnium oxide film experiences significant mechanical stress that lowers the temperature of phase transition from the high-temperature parent nonpolar cubic *P*m3m and tetragonal *P*4_2_/nmc phases to the polar orthorhombic phase [2,41,42,43]. In this work, TCEs of the substrate and functional materials are equal to the following values for Si substrate: 2.6 × 10^−6^/°C [41], for HfO_2_: 8…10 × 10^−6^/°C [44], for W: 4.5 × 10^−6^/°C [42], for WO_3_: 8…15 × 10^−6^/°C [45,46,47]. Therefore, during annealing, the Si substrate induces mechanical stress in HZO and facilitates the formation of the ferroelectric phase. The W bottom electrode leads to the same effect, whereas the WO_x_ interlayer has the same TCE as HZO or even larger and thus suppresses the mechanical stress from the substrate. This suppression is greater the thicker the WO_x_ interlayer is. As can be seen from the XRD results, this effect does not lead to the formation of a non-ferroelectric monoclinic phase but may be the cause of the different texture of the ferroelectric phase grains, which is discussed above.

Third, the different layer thickness explains why the dielectric permittivity of samples No. 1 and 2 differs. It can be seen from Figure 1h that the dielectric permittivity of sample No. 1 is larger than that of sample No. 2. This could be due to the anisotropy of the dielectric permittivity; however, theoretical calculations show that it is small in hafnia. We assume that the difference is just due to the different thickness of the WO_x_ interlayer. Using the values of the total dielectric permittivity (Figure 1h) and thicknesses of HZO and WO_x_ layers, one could calculate their dielectric permittivity: 29 and 18 for the HZO and WO_x_ layers, respectively.

Finally, it is worth noting that a reasonably thicker WO_x_ interlayer results in smaller leakage currents of sample No. 2 compared to sample No. 1 (Figure 1d).

Thus, the WO_x_ interlayer affects the ferroelectric and dielectric properties of the film but does not explain the significant broadening of the switching time distribution nor, ultimately, the polarization switching retardation.

One of the most natural causes could be the different distribution of the electric field in HZO film. It can originate from the different roughness of the bottom electrode surface. Indeed, the developed surface leads to large variations in the electric field in the film through the effect of strengthening/weakening the field by bulges and pits. As discussed above, analysis of HRTEM images shows that the tungsten surface roughness is very similar for both samples. However, this method suffers from poor statistics, so we investigate the W/HZO surface using AFM and SEM.

The AFM and SEM images of the W/HZO surface look similar for both samples (Figure 6a,b). Quantitative analysis of the images yields the following results. The surface roughness obtained from AFM topography is 0.69 and 0.75 nm for samples No. 1 and 2, respectively; that is, they are almost equal to each other. These values differ (they are smaller) from the roughness that is obtained by the analysis of TEM images. This is natural, since in AFM and TEM areas of a very different size are analyzed. In addition, in AFM, the topography of the HZO surface is measured, while in TEM the cross-section of the W/HZO interface is investigated. Nevertheless, both the AFM and TEM results indicate that the roughness of the W bottom electrode is similar for both samples. Therefore, the electric field distributions due to the roughness are the same and do not affect the switching speed. 

Another factor that can affect the switching speed both directly and through the electric field is the grain size. Indeed, using phase-field modeling, it has been shown that the grain pattern (e.g., grain size and orientation) and the existence of grain boundaries affect the polarization switching or domain evolution dynamics in ferroelectric perovskite films [19,20]. However, the HZO grain size distributions acquired from the AFM and SEM images indicate that these distributions are very similar for both samples (Figure 6c–f). According to AFM results, in both samples, the largest number of grains are about 20 nm in size (Figure 6c), while the largest area of the HZO film is occupied by grains about 60–70 nm in size (Figure 6d). According to SEM results, the grain sizes are smaller: the largest number of grains are about 12 nm in size (Figure 6e), while the largest area of the HZO film is occupied by grains about 20 nm in size (Figure 6f). Since the SEM has a higher spatial resolution than the AFM, it gives a more correct grain size. In any case, both microscopies show that the grain size is almost the same for both samples.

Thus, we reveal that switching speed is not affected by the following properties of the samples: structural phase composition, film structure, the degree of its (non-)stoichiometry, W surface roughness, and HZO grain size. According to these parameters, the samples are very similar, and their only difference is a small difference in the thickness of the WO_x_ interlayer.

To gain further understanding, the domain structure of both samples and their transformation during polarization switching were investigated by means of the EBIC technique [24]. In this technique, the domain structure is mapped in situ in the ferroelectric capacitor, and simultaneously the morphological image of the top TiN electrode surface is acquired using back-scattered electrons mode. The polarization vector is visualized through mapping of the electron-beam induced current, which is defined by the redistribution of the generated electrons and holes in the local field of the spontaneous polarization (depolarization field). 

It is remarkable that the morphology of the top TiN electrode surface is similar for both samples (Figure 7a), whereas the domain structures drastically differ (Figure 7b). The main difference consists of the size of the domains. Specifically, the domains in the structure of type No. 1 are about four times larger than those in the structure of type No. 2. The similarity of the top electrode morphology indicates that the reason for the different domain size is not due to structural factors. Taking into account the possible reasons ruled out by the studies above, the only explanation for the observed phenomenon is the different densities of charged defects in the bulk of the HZO film. Usually, charged defects in hafnium oxide films are associated with charged oxygen vacancies formed due to oxidation–reduction chemical reaction with electrode materials [39,40]. As shown above, the HZO films are stoichiometric, i.e., the defect density is not too large. It was previously shown to be 10^18^–10^19^ cm^−3^ in HZO [23]. The change in defect density within an order in this range is difficult to detect, but this change can manifest itself indirectly.

Point charged defects are known to modify the local field in ferroelectric and dielectric materials. Therefore, the field distribution in the HZO film is indeed blurred. However, this is not due to differences in roughness or grain pattern, but due to a direct modification of the electric field. The broadening of the field distribution within the HZO film is the direct cause of the broadening of the switching time distribution and, given the different thicknesses of the interlayer, the slowing down of the switching speed.

To verify the assumption on different densities of charged defects in the two types of structures, we measures the capacitance transient curves using capacitance transient spectroscopy, which is a special case of impedance spectroscopy [48]. The transient curves of capacitance relaxation were measured after saturation of the charge traps by applying a long pulse of high amplitude (3 V/5 s). Figure 2d shows that for both samples, the rate of change in capacitance increases with increasing temperature. Since the capacitance depends on the charge within the HZO film and the potential across it, the increase in relaxation rate indicates an acceleration in charge emission with increasing temperature, which is natural [48]. At each individual temperature of 40 and 80 °C, a higher rate of relative capacitance relaxation was observed for the structure of type No. 2. In general, the rate of capacitance relaxation depends on several parameters, including traps density, traps energy, and the thickness of the interface dielectric layer. In particular, a higher rate indicates a higher density of charged traps [48,49], i.e., defects. Thus, the results of capacitance transient spectroscopy confirm the higher density of charged defects in the sample with a thicker WO_x_ interlayer (type No. 2).

It should be noted that the oxygen vacancy density could affect not only the switching speed but also the structural properties of hafnium oxide and thus the magnitude of the remanent polarization. Several works on this issue are known, but the role of this phenomenon is still unclear, and additional studies are required to clarify it.

There are several possible reasons for the variation in defect density from sample to sample. All of them are related to the sample-fabrication technology, including technological processes of the bottom electrode deposition, HZO growth, treatment their interface, and annealing conditions. Different dielectric interlayer thicknesses indicate different degrees of oxidation of the electrode due to the reduction of hafnium-zirconium oxide, which is accompanied by the formation of oxygen vacancies in the ferroelectric. The larger interlayer thickness in the structures of type No. 2 indicate a higher density of charged oxygen vacancies, which is consistent with all the results obtained.

## 4. Conclusions

We present a comprehensive study of possible contributions to the polarization switching speed in a polycrystalline ferroelectric HZO film. For this purpose, we fabricated two types of samples consisting of identical layers, but with a modified electrode interface. They demonstrated different dielectric and ferroelectric properties, including significantly different polarization switching speeds. The structural (including microstructural), chemical, morphological, and local ferroelectric properties of the samples were then investigated in detail by a number of advanced techniques, including EBIC, HAXPES, HRTEM, EDX, XRD. We revealed the way in which the thickness of the dielectric interlayer affects the measured remanent polarization, coercive voltage, dielectric permittivity, and leakage current of HZO film. However, the most dramatic effect was caused by the variation in the density of charged defects, which strongly affected the switching speed. This phenomenon originates from the effect of the charge distribution on the electric field inhomogeneity in the ferroelectric film. In other words, the charged defects are pinning centers that delay the domain wall propagation during polarization switching. A large density of defects leads to a slower switching speed.

Polarization switching kinetics determine the operating frequency and operating voltages of the memory chip and in-memory computing devices and, thus, addressing this challenge is crucial for the implementation of non-volatile ferroelectric memory and innovative computing systems. 

In addition, the results may be useful for applications based on the use of multiple states given by different domain configurations (neuromorphic devices, ferroelectric transistors, and ferroelectric tunnel junctions). As shown earlier, when devices are ultrascaled, the informative signal starts to be affected by individual domains [50]. In this work, we demonstrate a way to tune the domain size and reveal the relationship between the domain size and the required operating frequency of these types of devices.

## Figures and Tables

**Figure 1 nanomaterials-13-03063-f001:**
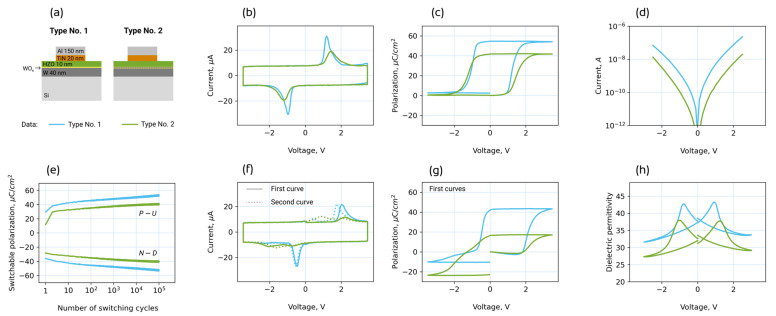
Ferroelectric and dielectric properties. (**a**) Sketch of two types of W/HZO/TiN structures, (**b**) switching *I-V* curves, (**c**) *P-V* curves, and (**d**) quasistatic *I-V* curves after the wake-up procedure. (**e**) Evolution of switchable polarization during the wake-up procedure. The designations *P − U* and *N − D* denote switchable polarization values measured by positive and negative pulses in the PUND-like technique. (**f**) First and second switching *I-V* curves and (**g**) first *P-V* curves; (**h**) *k-V* curves after the wake-up procedure. In (**c**,**g**), it can be noted that the *P-V* curves start from zero polarization, which is due to the use of the PUND-like technique [25].

**Figure 2 nanomaterials-13-03063-f002:**
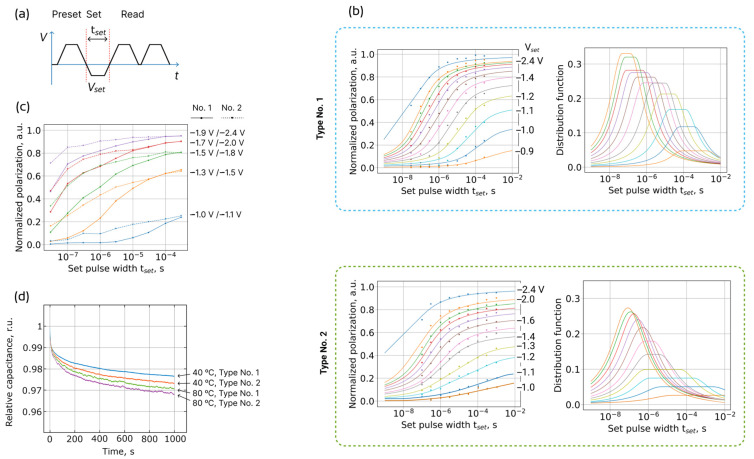
Polarization switching kinetics. (**a**) The sequence of pulses used to obtain data on switching of the polarization depending on the amplitude and duration of the set pulse. (**b**) Polarization switching kinetics for two types of structures: datasets (dots) and fitting by the NLS model (lines). Different colors correspond to different *V*_set_, some of which are listed in the legend. (**c**) Comparison of experimental switching speed, (**d**) capacitance transient curves for structures of both types measured at 40 and 80 °C at zero DC voltage after applying voltage pulse 3 V/5 s. In (**d**), the initial 100 ns are removed to skip the exponent component of the *RC* circuit discharging.

**Figure 3 nanomaterials-13-03063-f003:**
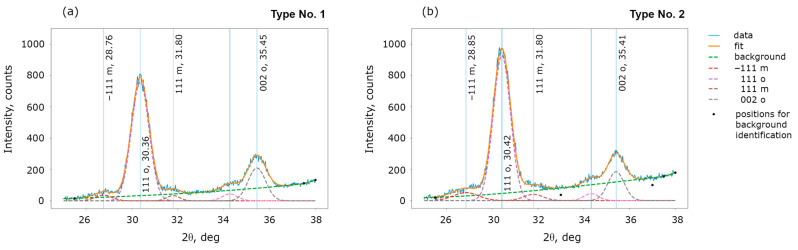
Comparison of the structural phase composition in the structures of type (**a**) No. 1 and (**b**) No. 2 by XRD patterns.

**Figure 4 nanomaterials-13-03063-f004:**
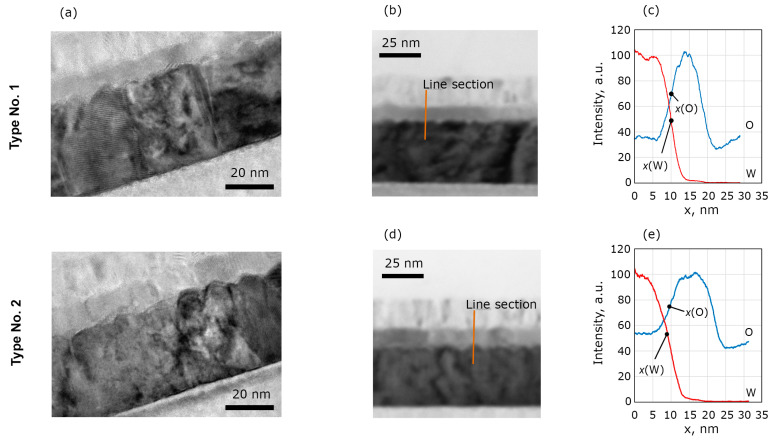
(**a**) HRTEM images of the structures of No. 1 and 2 cross-sections. (**c**) EDX profiles of O and W obtained in the region (**b**) of the structure of type No. 1. (**e**) EDX profiles of O and W obtained in the region (**d**) of the structure of type No. 2. The profiles are treated with *k*-nearest neighbor smoothing.

**Figure 5 nanomaterials-13-03063-f005:**
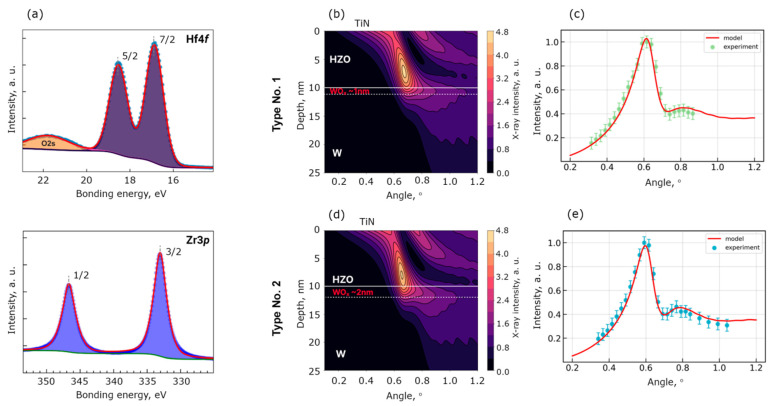
Results of HAXPES study. (**a**) The Hf4f and Zr3p core-level lines for sample No. 1. (**b**,**d**) corresponded to best-fit X-ray intensity distribution maps across a W/HZO structure for samples No. 1 and 2. (**c**,**e**) Fitted Hf4f7/2 angular intensity distribution of samples No. 1 and 2.

**Figure 6 nanomaterials-13-03063-f006:**
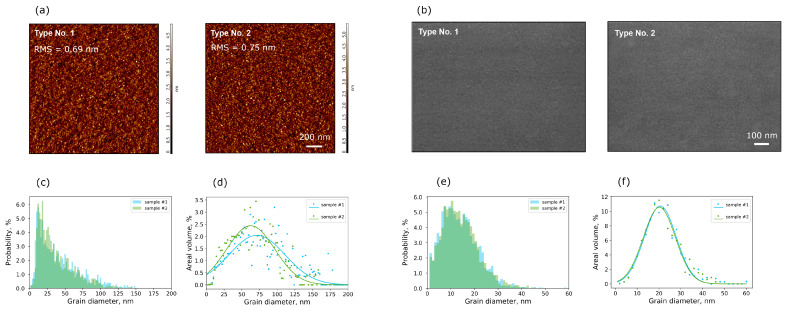
Images of the W/HZO surface. (**a**) AFM topography, (**b**) SEM images. Distribution of (**c**) number grain size and (**d**) their area fraction acquired from AFM topography maps. Distribution of (**e**) number grain size and (**f**) their area fraction acquired from SEM images.

**Figure 7 nanomaterials-13-03063-f007:**
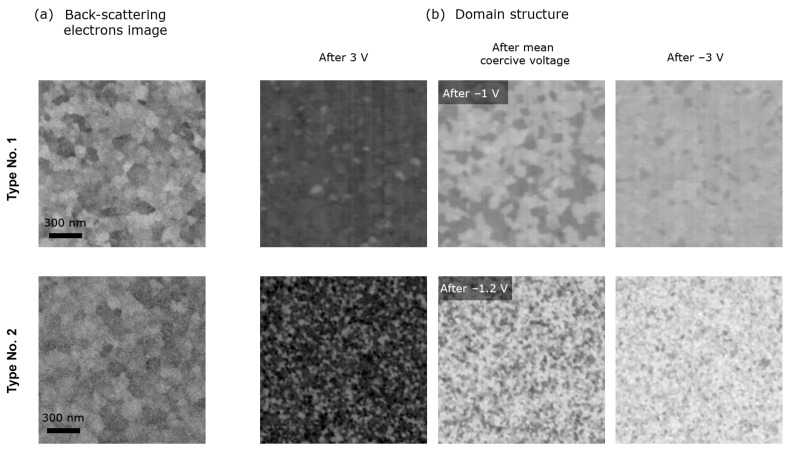
Domain structures. (**a**) Back-scattering electrons images associated with the morphology of the top TiN electrode. (**b**) EBIC maps measured in the same region after the applying different voltage pulses. They are acquired at the same imaging parameters, including magnification.

## Data Availability

Data are contained within the article.

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
