# Peer review of "Effect of Domain Structure and Dielectric Interlayer on Switching Speed of Ferroelectric Hf_0.5_Zr_0.5_O_2_ Film"

_nanomaterials, 2023, doi:10.3390/nano13233063_

Round 1
Reviewer 1 Report
Comments and Suggestions for Authors
This manuscript reveals that the domain structure and the dielectric layer formed at the electrode interface contribute to the polarization switching speed of HZO film. This work is helpful to the investigation of non-volatile ferroelectric memory. Nevertheless, several problems need to be thought them over and to be worked out.
1. The authors suggest that the polydomain structure of type No. 2 is caused by the charged oxygen vacancies formed at both bottom and top electrode interfaces during annealing. How do the oxygen vacancies induce the polydomain structure? It is not clear whether or not the oxygen vacancies exist in the interfaces.
2. Why does a high ratio of intensities of peaks 002 and 111 cause the larger remanent polarization? Do the oxygen vacancies, thickness and density of films have an effect on the remanent polarization?
3. The average h of No. 1 and No. 2 is obvious different. What is the reason? The EDX profiles of O and W of the structure No. 1 should be shown in the paper.
Comments on the Quality of English Language
For the English of the manuscript, many revisions (e.g. typos, definite articles, plurals, inappropriate words) are needed.
Author Response
- The authors suggest that the polydomain structure of type No. 2 is caused by the charged oxygen vacancies formed at both bottom and top electrode interfaces during annealing. How do the oxygen vacancies induce the polydomain structure? It is not clear whether or not the oxygen vacancies exist in the interfaces.
We agree that several hypotheses are known to explain the reasons for the formation of the polydomain state in as-prepared ferroelectric HfO2 films. The contribution of oxygen vacancies is one of the most common hypotheses. We have added a comment on this issue.
In the revised manuscript (p. 5-6): “The polydomain structure that is peculiar to the structure of type No. 2 is caused by the charged oxygen vacancies formed at both bottom and top electrode interfaces during annealing because of redox reactions with the electrode materials [3]. Depending on the mutual distribution of oxygen vacancies at the two interfaces during crystallization into the ferroelectric phase, their sum electric field locally switches the HZO film to either upward or downward polarization. Immediately after local polarization switching, the emergence of built-in electric fields and their gradual increase with time are observed. This phenomenon is caused by the redistribution of non-ferroelectric charges under the action of the depolarization field, which is aligned with the polarization vector. As a result, two opposite built-in fields appear in the domains of two types, and they manifest themselves in the splitting of the I-V curve and the pinching of the P-V curve.”
- Why does a high ratio of intensities of peaks 002 and 111 cause the larger remanent polarization? Do the oxygen vacancies, thickness and density of films have an effect on the remanent polarization?
We are grateful for this comment. We have added discussions about the intensity ratio of the 002 and 111 peaks and the role of oxygen vacancies in the measured remanent polarization. Since the HZO film was grown in the same ALD process for the both types of samples, discussion of the role of film thickness and film density seems unnecessary.
In the revised manuscript (p. 8): “The only difference in the XRD patterns is the ratio of intensities of peaks 002 and 111 of the orthorhombic ferroelectric phase, which is larger for film No. 1. Peak 002 corresponds to the vertical orientation of the polar axis, while peak 111 corresponds to the polar axis oriented at an angle of 57.7° to the film plane. As was shown earlier [37], a larger ratio of the intensities of peaks 002 and 111 correlates with a larger measured remanent polarization. In this work, a correlation between a larger ratio of peak intensities and a larger residual polarization is also observed (Figure 1c).”
In the revised manuscript (p. 14): “It should be noted that the oxygen vacancy density can affect not only the switching speed but also the structural properties of hafnium oxide and thus the magnitude of the remanent polarization. Several works on this issue are known, but the role of this phenomenon is still unclear, and additional studies are required to clarify it.”
- The average h of No. 1 and No. 2 is obvious different. What is the reason? The EDX profiles of O and W of the structure No. 1 should be shown in the paper.
An example of the EDX profiles of W and O for the structure of type No. 1 has been added. The difference in h values has been commented.
In the revised manuscript (p. 9): “Average values of h = – 0.46 nm (RMS of 0.6 nm) and h = – 0.25 nm (RMS of 1 nm) are found for structures of type No. 1 and 2, respectively. Thus, the difference in h values is small and can be attributed to the modification of the W interface of the structures No. 1 prior to HZO growth”.
Reviewer 2 Report
Comments and Suggestions for Authors
(1) Please explain the reason for the asymmetry of the P-V curve in Figure 1f.
(2) Domain structure has an impact on polarization switching speed, and it is necessary to supplement PFM testing of two devices to observe the domain reversal behavior of the devices.
(3) On page 5, line 215, "This indicates the presence of oxygen vacancies at one of the electronic interfaces..." It is recommended to perform ion etching on the sample and use XPS to investigate the oxygen vacancy content at the device interface.
(4) Figures 3 (a) and (b) cannot clearly distinguish the WO3 layer. It is recommended to add selected area electron diffraction of two devices at the interface to analyze the layering at different interfaces clearly.
(5) In line 313, 'several EDX profiles (Figure 3c, d) are observed and analyzed for each cross section', Figure 3d was not found in the article, It is crucial for proving the author's point of view.
(6) Please add magnification/ruler in Figure 6.
Author Response
- Please explain the reason for the asymmetry of the P-V curve in Figure 1f.
We have added a comment on this issue.
In the revised manuscript (p. 5-6): “In order to get an initial idea of the reasons for the difference in ferroelectric proper-ties, the switching I-V and P-V curves of the as-prepared structures can be analyzed. The very first two curves reflect the state of the as-prepared structure, i.e. whether the HZO film becomes mono- or polydomain right after crystallization. Indeed, obvious splitting of the switching I–V curve and related pinched P–V curve observed for the structure of type No. 2 (Figure 1f,g) indicates the existence of two opposite internal built-in fields, which are sig-natures of the polydomain structure in the as-prepared film [32, 33]. The polydomain structure that is peculiar to the structure of type No. 2 is caused by the charged oxygen vacancies formed at both bottom and top electrode interfaces during annealing because of redox reactions with the electrode materials. Depending on the mutual distribution of oxygen vacancies at the two interfaces during crystallization into the ferroelectric phase, their sum electric field locally switches the HZO film to either upward or downward polarization. Immediately after local polarization switching, the emergence of built-in electric fields and their gradual increase with time are observed. This phenomenon is caused by the redistribution of non-ferroelectric charges under the action of the depolarization field, which is aligned with the polarization vector. As a result, two opposite built-in fields appear in the domains of two types, and they manifest themselves in the splitting of the I-V curve and the pinching of the P-V curve.
For the structure of type No. 1, splitting of the switching I–V curve and pinching of P–V curve are much smaller. Instead, the P-V curve is shifted towards positive voltages, in-dicating a nearly monodomain state of the as-prepared structure and hence a unidirec-tional internal field built-in in the HZO film. This is probably due to the absence of oxygen vacancies at one of the electrode interfaces, i.e., the chemical inertness of one of the elec-trodes. A possible reason could be the existence of a chemically inert dielectric layer WO3 formed at the lower interface before the growth of the HZO film.”
- Domain structure has an impact on polarization switching speed, and it is necessary to supplement PFM testing of two devices to observe the domain reversal behavior of the devices.
We agree that the PFM is the most widely known tool to study the domain reversal behavior. However, it has limited spatial resolution. In previous work (ref. 24), we proposed an advanced technique of the domain mapping, which based on mapping of electron-beam-induced current. We compared the EBIC technique with the PFM technique and found that it has much better resolution. In current work, we employ this advantage of the EBIC technique because the domains in the structure of type No. 2 are very small.
- On page 5, line 215, "This indicates the presence of oxygen vacancies at one of the electronic interfaces..." It is recommended to perform ion etching on the sample and use XPS to investigate the oxygen vacancy content at the device interface.
This remark of the referee means that we did not clearly articulate in the manuscript how we can elucidate the presence of oxygen vacancies in the bulk of HZO and/or at interfaces. The point is that neutral oxygen vacancies at the concentrations of 1018-1020 cm-3 are below the sensitivity of XPS. However, charged oxygen vacancies present in a dielectric at such concentrations give rise to the potential distribution across the layer, thus affecting the energy of photoelectrons and ultimately the core-level spectra of dielectric constituents. Therefore, the observed broadening and shifting of dielectric core-level spectra can be attributed charged defects, and that is how we can reconstruct their location. We have added clarification on this issue while describing HAXPES methodology on p. 4. As for ion etching on the sample to reveal depth dependent peculiarities, the very idea of NTR-HAXPES is to perform depth analysis non-destructively to avoid any ion etching which is known to affect chemical composition/electronic properties in the bulk and at interfaces. We have rewritten the whole section related to HAXPES methodology on p. 4 to make it more clear why this technique in NTR mode was used in this study.
- Figures 3 (a) and (b) cannot clearly distinguish the WO3 layer. It is recommended to add selected area electron diffraction of two devices at the interface to analyze the layering at different interfaces clearly.
HRTEM images do not reveal nanocrystals between W and HZO layers. At the same time, amorphous WO3 can not be detected on the selected area diffraction because of the nature of the FIB prepared cross-section – it has some volume of amorphous material under the side surfaces of the cross-section. Furthermore, in our microscope, minimal selected area is of 200 nm in diameter; therefore, the selected area would contain amorphous SiO2 under W electrode or amorphous carbon protective material above the top TiN electrode.
- In line 313, 'several EDX profiles (Figure 3c, d) are observed and analyzed for each cross section', Figure 3d was not found in the article, It is crucial for proving the author's point of view.
We apologize for the confusion, it was a typo. We should have written '(Figure 3b, c)'. In the revised manuscript, we additionally have provided an example of EDX profiles of W and O (Figure 4b-e).
- Please add magnification/ruler in Figure 6.
We are grateful for this comment. Scale bars have been added to Figure 6 (Figure 7 in the revised manuscript).
Reviewer 3 Report
Comments and Suggestions for Authors
The authors report a study on engineering ferroelectric/electrode heterostructure (and their dielectric environment) to modify the switching speed and ferroelectric properties. Several characterization techniques are used including EBIC, HAXPES, TEM, XRD… providing a consistent set of data. Modification of various ferroelectric figures of merits (polarization, domain kinetics, coercive voltage, dielectric permittivity, leakage current) is reported depending on the ferroelectric interface, which is explained and modelized as the variation in the density of charged defects, that play the role of pinning centers that tune the domain wall dynamics.
I found the paper and results interesting, providing new information on the interfacial engineering of ferroelectric film and ferroelectric switching performances.
I provide a list of questions and comments I would like the authors to consider :
1) I found several typos in the main text (for example : page 10 “I can originate from” instead of 3It can..” ; in the conclusion “significsntly” instead of “significantly”). The authors should perform proofreading of the manuscript.
2) Page 3, line 103 : can the author better specify what is referred as “wake up procedure” and its goal.
3) Figure 1 : The legend of the figures is not clear. The author should specify what is blue “Type No1” and green “Type n2” for figure 1a-f. So that the reader can have access directly to the information. For this, the authors should include a sketch of the stack of Type 1 and Type 2 device.
4) Figure 1.a : can the authors comment why the P((V) loop is pinned on the positive polarity and does not show negative polarization after wake-up procedure ?
5) Did the authors perform sub-coercive field P(V) loop of various Voltage window size ( 0 to +0.5 to -0.5 to +1 to -1 etc….) ? This could provide additional insight into the FE domain switching properties and multilevel encoding capabilities?
6) Can the author perform impedance spectroscopy and frequency-dependent capacitance measurement to provide additional experimental evidence of the proposed two-capacitor model discussed page 10-11 ? This would also enable quantifying the trap state/charged defects density?
7) In order to make the paper even more appealing for a larger audience, the authors should consider putting their work in perspective with timely issues in ferroelectric devices and beyond Moore applications. Here are some suggestions that the author should consider to adapt/include either in the introduction or conclusion :
i) The control of domains in ferroelectric thin film and theur switching speed and distribution is of highest interest for in-memory computing, covering large range of applications going from neuromorphism (see for example https://doi.org/10.1103/PhysRevLett.126.027602 ) and in-memory ferroelectric logics (see for example : https://doi.org/10.1021/acsnano.3c07952) . The authors should introduce this thematic in the introduction, in order to make the paper matching a larger audience, and to broaden the perspectives.
ii) The author should also consider to discuss about the interest of taking advantage of controlling ferroelelctric domain dynamics to encode multiple state for neuromorphic appplications. This may for example be used to trigger multiple state in order emulate synaptic potentiation/depression functionality ( see implementation in thin film , HfZO ) following multi-domain NLS transient model.
(the proposed references are only indicative and could be completed with additional ones when relevant)
Comments on the Quality of English Language
I found several typos in the main text (for example : page 10 “I can originate from” instead of 3It can..” ; in the conclusion “significsntly” instead of “significantly”). The authors should perform proofreading of the manuscript.
Author Response
- I found several typos in the main text (for example : page 10 “I can originate from” instead of 3It can..” ; in the conclusion “significsntly” instead of “significantly”). The authors should perform proofreading of the manuscript.
We are grateful for this comment. The manuscript has been proofread.
- Page 3, line 103 : can the author better specify what is referred as “wake up procedure” and its goal.
We have added a comment on this issue.
In the revised manuscript (p. 6): “During multiple switching by bipolar voltage pulses (so called wake-up procedure), charged oxygen vacancies redistribute from the electrode interfaces into the film bulk. As a result, the built-in fields disappear, the I-V and P-V curves become non-split and open respectively, the switchable polarization increases (Figure 1e) and the ferroelectric properties stabilize [25]. After the wake-up procedure, structures of both types show large remanent polarization and small coercive voltages, i.e. by these parameters, they are suitable for the implementation of ferroelectric memory.”
- Figure 1 : The legend of the figures is not clear. The author should specify what is blue “Type No1” and green “Type n2” for figure 1a-f. So that the reader can have access directly to the information. For this, the authors should include a sketch of the stack of Type 1 and Type 2 device.
We are grateful for this advice. A sketch has been added to Figure 1.
- Figure 1.a: can the authors comment why the P(V) loop is pinned on the positive polarity and does not show negative polarization after wake-up procedure?
In our previous work, we have discussed in detail the PUND-like method that leads to such shape of P-V curves. Therefore, we have noted this issue and provided a reference to the previous work.
In the revised manuscript (caption to Figure 1): “In (c) and (g), it can be noted that the P-V curves start from zero polarization, which is due to the use of the PUND-like technique [25].”
- Did the authors perform sub-coercive field P(V) loop of various Voltage window size ( 0 to +0.5 to -0.5 to +1 to -1 etc….) ? This could provide additional insight into the FE domain switching properties and multilevel encoding capabilities?
Thank you for the interesting idea. In fact, the polarization switching kinetics data is actually measured using sub-coercive field P-V loops. The amplitude of the set pulse in the pulse sequence in Fig. 2a varies over a wide voltage range. Therefore, some values of the measured polarization correspond to the saturated polarization of sub-coercive field P-V loops.
- Can the author perform impedance spectroscopy and frequency-dependent capacitance measurement to provide additional experimental evidence of the proposed two-capacitor model discussed page 10-11? This would also enable quantifying the trap state/charged defects density?
We are grateful for this advice and stimulus to improve our work. We have added the results of capacitive transient spectroscopy. They also indicate a larger concentration of defects in structures with smaller domain size.
In the revised manuscript, Figure 2d and an appropriate text (p. 13-14): “To verify the assumption on different density of charged defects in the two types of structures, we measure the capacitance transient curves using capacitance transient spectroscopy, which is a special case of impedance spectroscopy [48]. The transient curves of capacitance relaxation are measured after saturation of the charge traps by applying a long pulse of high amplitude (3 V/5 s). Figure 2d shows that for both samples, the rate of change of capacitance increases with increasing temperature. Since the capacitance depends on the charge within the HZO film and the potential across it, the increase in relaxation rate indicates an acceleration of charge emission with increasing temperature, which is natural [48]. At each individual temperatures of 40 and 80 °C, a higher rate of relative capacitance relaxation is observed for the structure of type No. 2. In general, the rate of capacitance relaxation depends on several parameters, including traps density, traps energy and the thickness of the interface dielectric layer. In particular, a higher rate indicates a higher density of charged traps [48, 49], i.e., defects. Thus, the results of capacitance transient spectroscopy confirm the higher density of charged defects in the sample with thicker WOx interlayer (type No. 2).”
- In order to make the paper even more appealing for a larger audience, the authors should consider putting their work in perspective with timely issues in ferroelectric devices and beyond Moore applications. Here are some suggestions that the author should consider to adapt/include either in the introduction or conclusion:
- i) The control of domains in ferroelectric thin film and their switching speed and distribution is of highest interest for in-memory computing, covering large range of applications going from neuromorphism (see for example https://doi.org/10.1103/PhysRevLett.126.027602 ) and in-memory ferroelectric logics (see for example : https://doi.org/10.1021/acsnano.3c07952) . The authors should introduce this thematic in the introduction, in order to make the paper matching a larger audience, and to broaden the perspectives.
- ii) The author should also consider to discuss about the interest of taking advantage of controlling ferroelelctric domain dynamics to encode multiple state for neuromorphic applications. This may for example be used to trigger multiple state in order emulate synaptic potentiation/depression functionality ( see implementation in thin film , HfZrO ) following multi-domain NLS transient model.
(the proposed references are only indicative and could be completed with additional ones when relevant)
We are grateful for this comment and stimulus to make our appealing for a wider audience. Appropriate discussions have been added to Introduction and Conclusions sections.
In the revised manuscript (p. 1-2): “Polarization switching speed and related domain dynamics are important characteristics for other applications of ferroelectric films as well, for example, for in-memory computing. This direction includes in-memory ferroelectric logics [5-7] and neuromorphic computing [8, 9]. Neuromorphism of ferroelectric structures stems from the multiple states provided by varying domain configurations. Different neuromorphic applications demand a wide range of operating frequencies – from hundred Hz for emulation of biological synapses [10-12] to THz for ultrafast processing of data and reduced energy consumption [13]. The design and implementation of such devices requires an understanding of both the intrinsic kinetics of polarization switching and the external mechanisms that lead to its degradation in real devices.”
In the revised manuscript (p. 14): “Polarization switching kinetics determines the operating frequency and operating voltages of the memory chip and in-memory computing devices and thus addressing of this challenge is crucial for the implementation of non-volatile ferroelectric memory and innovative computing systems.
In addition, the results may be useful for applications based on the use of multiple states given by different domain configurations (neuromorphic devices, ferroelectric tran-sistors and ferroelectric tunnel junctions). As shown earlier, when devices are ultrascaled, the informative signal starts to be affected by individual domains [50]. In this work, we demonstrate a way to tune the domain size and reveal the relationship between the do-main size and the required operating frequency of these types of devices.”
Round 2
Reviewer 2 Report
Comments and Suggestions for Authors
The author solved the problem we raised.
Reviewer 3 Report
Comments and Suggestions for Authors
The authors provided the mandatory changes. The article can be published in Nanomaterials.